# Investigation of Microbiological Quality Changes of Roof-Harvested Rainwater Stored in the Tanks

Monika Zdeb *, Justyna Zamorska, Dorota Papciak and Agata Skwarczyńska-Wojsa

Department of Water Protection and Purification, Rzeszów University of Technology, 35-959 Rzeszów, Poland; jzamor@prz.edu.pl (J.Z.); dpapciak@prz.edu.pl (D.P.); askwarczynska@prz.edu.pl (A.S.-W.)
* Correspondence: mzdeb@prz.edu.pl; Tel.: +48-017-865-1316

**Abstract:** Rainwater has been found to be a valuable source of drinking water in Europe, especially in such crisis situations as those caused by contamination of water uptake into water supply systems, large-scale floods or terrorist attacks (e.g., biological weapons). The microbiological quality of water plays a significant role, which is directly related to the potential health risks associated with harvested rainwater (including rainwater stored in the tanks). Microbial contamination is commonly found in rainwater. However, in the literature, detailed results of qualitative and quantitative microbiological assessments are sparse and remain unexplored. Therefore, the aim of this study was to investigate and analyze changes in the microbiological quality of roof-harvested rainwater stored in the tanks, depending on the collection conditions (type of roof surface), storage duration and season. Authors elucidate that conditions such as storage duration, the season in which rainwater is collected, the roof-like surface types and morphology of the catchment area highly affect rainwater quality. This study showed that rainwater harvested from a galvanized steel sheet roof had the best microbial quality, regarding the lowest number of bacteria, while rainwater from a flat roof covered with epoxy resin was the worst. Further, it was detected that rainwater collected in autumn and spring obtained the best microbiological quality. Moreover, a decrease in the number of bacteria was observed in correlation to storage duration. The water became sanitary safe after six weeks of storage at 12 °C. Its use for purposes requiring drinking water quality before six weeks of storage required disinfection.

**Keywords:** microbiological quality; roof harvested rainwater; storage tanks; ATP concentration

## 1. Introduction

The use of rainwater as drinking water is common in countries where groundwater and surface water resources are scarce [1,2].

Moreover, Rain Water Harvested Systems (RWHS) are an affordable and sustainable alternative, especially in the areas with dispersed housing, where the costs of building traditional water supply systems are very high [3,4].

Rainwater collection techniques are used around the world to support drinking water supply [5,6], manage rainwater and reduce flood risk by reducing the volume of water flow in storm sewer systems and also as efficient and ecological functionality for buildings [7,8].

The main assumptions of water management are embraced in the Directive 2000/60/EC of the European Parliament and of the Council establishing a framework for the Community action in the field of water policy Water Framework Directive (WFD). It requires the rational and sustainable use of water resources in citizens' demand for water, agriculture and industry, to achieve the environmental objectives under the directive by maintaining and protecting the environment [9].

In order to reduce the runoff of rainwater out of its catchment area and to reduce the exploitation of surface and groundwater, the RWHS for collection and storage of rain can be used [10,11]. Such systems may diminish the effects of temporal and spatial variability of rainfall, such as drought and floods [12], improve water retention in the environment, and help manage and use rainwater for basic human needs or small-scale production [13,14].

RWHS are quite common in many countries, mainly thanks to considerable economic benefits, such as saving tap water, lower charge for its supply and for sewage disposal [15–17].

In European conditions, Council Directive 98/83/EC from 3 November 1998 on the quality of water intended for human consumption defined that "water intended for human consumption" shall mean: "all water either in its original state or after treatment, intended for drinking, cooking, food preparation or other domestic purposes, regardless of its origin and whether it is supplied from a distribution network, from a tanker, or in bottles or containers", which means that rainwater can be used as a valuable source of drinking water in times of crisis caused by water supply system failures, large-scale floods or terrorist attacks, for example, by using biological weapons. Moreover, the aforementioned directive defines parameters and parametric values, including the acceptable values of indicators (*Escherichia coli*) were also defined for situations in which it is impossible to obtain and maintain water of the highest microbiological quality [18].

Rainwater may be contaminated with microorganisms already at the stage of precipitation formation, during the runoff from the surface from which it is collected or at rainwater harvesting systems [19].

Rainwater collected directly from roofs indicates good quality compared to that collected from surface runoff, for example, from roads, parking lots or green areas [20]. The main factors affecting the quality of roof drainage include the following three aspects: (1) roof features (e.g., type, age, roughness), (2) precipitation characteristics (e.g., duration and intensity of rain fall), (3) environmental features (e.g., seasonal changes, terrain, wind direction, vegetation) [21,22].

The microbiological aspect in the case of using water plays a significant role and directly determines sanitary risk [23].

However, the use of rainwater for domestic and other purposes enforces an assessment of their physicochemical and microbiological quality [24]. Microbial pollution of rainwater is common. However, detailed results of microbiological qualitative and quantitative assessment in the literature are relatively rare and remain unexplored [25–27].

However, sustainable management of rainwater involves not only its collection but also storage. During the storage of rainwater, secondary contamination or changes in quality of water may occur, therefore it is important to indicate the parameters enabling a quick assessment of suitability, for example, drinking or hygienic purposes.

The present work demonstrates an assay of changes in the microbiological quality of collected rainwater stored in tanks depending on the type of roofing cover, storage duration and season.

Therefore, the following crucial evaluations were presented for the first time in this study:

- To establish optimal conditions for rainwater storage process to obtain suitable microbiological quality, intended for household use;
- The correlation between physicochemical and bacteriological parameters, which would enable a quick assessment of the suitability of rainwater for a specific purpose, (e.g., potable water);
- Methodology for fast microbiological stored rainwater quality determining (especially in conditions of urgent need as a source of clean water).

The implementation of these findings may increase the safety of rainwater use also as a source of drinking water in crisis conditions, reduce the costs of collecting and using rainwater, and reduce the maintenance frequency of rainwater tanks.

## 2. Materials and Methods

### 2.1. Research Area

The rainwater harvesting area was located in the south-eastern part of Poland, near the city of Rzeszow (Figure 1). The collection of rainwater samples was carried out in a non-industrialized area in the estate of single-family houses. The region was characterized

as having a mountain climate (of the lowland, valley and mountain types), where westerly winds prevail.

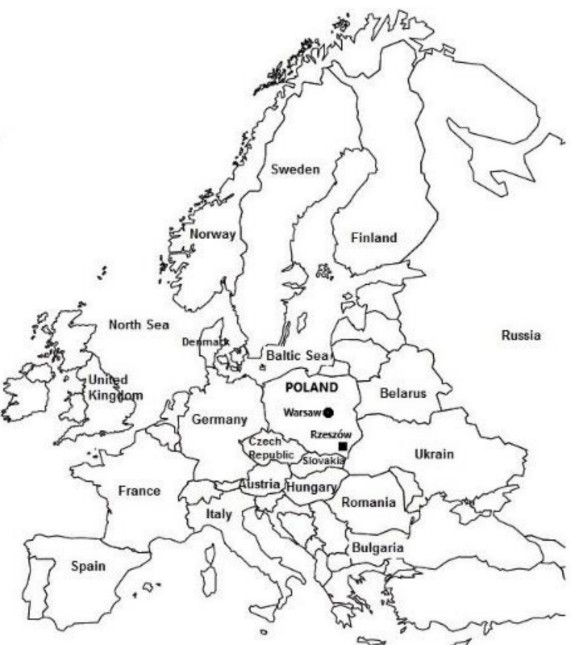

**Figure 1.** Location of Rzeszow—rainwater sampling area [28].

The area within 3 km from the samples collection point is characterized by low particulate matter emissions (no industrial plants) and organic emissions (only grass mowing). The closest communication route was a road with low traffic, located 0.3 km from the sampling point, and the national road (E4) was 5.5 km to the south. Moreover, the wastewater treatment plant was located northwest at a the distance 3.5 km away from the sampling point, and a combined heat and power plant was 5 km away. The area was covered with low shrub vegetation and grasses. The nearest stands and farmlands were located about 1 km west of the buildings.

*2.2. Rainwater Collection*

Rainwater was collected in spring (spr), summer (sum) and autumn (aut) and then stored. Samples were collected during selected rainy episodes according to rainfall intensity. After rejecting the so-called first runoff [28], samples were stored in the tanks. Rainwater was collected in disinfected, 30-L high-density polyethylene (HD-PE—100) canisters from roof gutters as runoff from roofs covered with: concrete tile (Con), ceramic tile (Cer), galvanized sheet (GS), epoxy resin (Epo) and directly from atmospheric precipitation (P) (Table 1). Each of the buildings was equipped with the same type of gutter system made of aluminum.

Rainwater collected directly from atmospheric precipitation was treated according to Polish regulations PN-ISO 5667-8:2003: "Water quality. Sampling. Guidelines for wet precipitations sampling". For this purpose, 10 high-density polyethylene (HD-PE—100) (40 × 40 cm) were placed 1.5 m above the ground level on a special stand construction. Finally, rainwater was collected in a 30 L polyethylene tank. In this way, the number of microorganisms only from the air was studied.

Disinfection of all canisters was carried out using a chemical method—ozonation. For this purpose, an ozonator generating 7 mg $O_3$/h was used. Ozonation was conducted for each tank for an hour.

The physicochemical quality of water was determined using a number of selected particular parameters that indicate the direct impact on the microbiological stability of

water (microbial growth), that is, biogenic substances C, N, P, as well as pH and water turbidity (Table 2).

**Table 1.** Characteristics of roof surfaces.

| Characteristic | Concrete Tile | Ceramic Tile | Epoxy Resin | Galvanized Steel Sheet |
|---|---|---|---|---|
| Location | 50°03′43.9″ N 22°06′53.6″ E | 50°03′45.2″ N 22°06′56.1″ E | 50°03′44.3″ N 22°06′53.5″ E | 50°03′51.5″ N 22°06′51.2″ E |
| Angle of inclination | 45° | 45° | 2° | 45° |
| Roof surface orientation | eastern | eastern | eastern | eastern |
| Components * | sand, Portland cement pigments—acrylic paint | clay, matte engage | Isocyanates, epoxy components, Pentamethyl piperidylsebacate, 3-isocyanatomethyl-3,5,5-trimethylcyclohexyl isocyanate, methyl 1,2,2,6,6-pentamethyl-4-piperidyl sebacate | iron, carbon, metals: chromium, nickel, manganese, tungsten, copper, molybdenum, titanium |

* details provided by the manufacturer.

**Table 2.** Initial rainwater parameters.

| | Season | Parameter | | | | | | |
|---|---|---|---|---|---|---|---|---|
| | | pH ±0.05 | Turbidity NTU * ±0.01 | mg $NNH_4^+$/L ±0.005 | mg $N\text{-}NO_2^-$/L ±0.005 | mg $N\text{-}NO_3^-$/L ±0.005 | mg $PO_4^-$/L ±0.005 | mg TOC **/L ±0.001 |
| P | spring summer autumn | 6.2 6.2 6.1 | 2.13 1.21 1.61 | 0.45 0.30 0.28 | 0.03 0.07 0.04 | 2.65 1.48 1.43 | 0.02 0.02 0.02 | 3.83 2.58 1.59 |
| Con | spring summer autumn | 6.5 6.5 7.4 | 4.15 4.79 3.21 | 0.29 0.32 0.31 | 0.06 0.02 0.07 | 1.30 2.26 2.43 | 0.02 0.04 0.02 | 2.13 4.87 2.91 |
| Cer | spring summer autumn | 6.0 6.4 7.1 | 5.72 4.74 1.97 | 0.28 0.28 0.21 | 0.05 0.04 0.03 | 3.12 1.84 1.46 | 0.02 0.03 0.02 | 3.51 2.85 3.32 |
| Epo | spring summer autumn | 6.3 6.5 6.8 | 6.11 4.44 4.27 | 0.20 0.30 0.30 | 0.05 0.08 0.05 | 3.43 1.87 2.56 | 0.03 0.04 0.02 | 6.17 4.33 4.25 |
| Gs | spring summer autumn | 6.1 6.0 5.8 | 3.17 3.13 1.12 | 0.47 0.15 0.09 | 0.04 0.08 0.07 | 2.22 0.81 1.13 | 0.01 0.02 0.02 | 1.89 3.01 2.61 |

* NTU—Nephelometric Turbidity Unit. ** TOC—Total Organic Carbon.

*2.3. The Rainwater Storage Conditions and Sampling Procedure*

All collected rainwater was kept at 12 °C in laboratory conditions with no access to light. The storage temperature and conditions were intended to stimulate the constant temperature and darkness characteristic of underground rainwater tanks [29].

All microbiological changes occurring in the rainwater during storage were monitored and verified within 3 months. Every 2 weeks, samples were taken for further analysis using a peristaltic pump and a sterile and disposable silicone tube. A sterile sampling tube was dipped to a depth of about 10 cm above the bottom of the canister and the peristaltic pump was turned on. Then the tube was moved to the surface of the water in the tank. The water sample collected was a mixed sample from almost the entire depth of the reservoir except for the bottom sediment, and overlying water from a few centimeters deep from the bottom of the tank. The obtained results of biogenic substances, pH and water turbidity of tested rainwater before and after storage process were compared and analyzed (Table 3).

**Table 3.** Rainwater parameters after six weeks of storage.

| | Season | Parameter | | | | | | |
|---|---|---|---|---|---|---|---|---|
| | | pH ±0.05 | Turbidity NTU ±0.01 | mg NNH$_4^+$/L ±0.005 | mg N-NO$_2^-$/L ±0.005 | mg N-NO$_3^-$/L ±0.005 | mg PO$_4^-$/L ±0.005 | mg TOC/L ±0.001 |
| P | spring | 6.4 | 2.07↓ | 0.15↓ | 0.01↓ | 0.55↓ | 0.01↓ | 1.49↓ |
| | summer | 6.2 | 1.05↓ | 0.11↓ | 0.04↓ | 0.58↓ | 0.01↓ | 1.65↓ |
| | autumn | 6.1 | 0.99↓ | 0.09↓ | 0.01↓ | 0.43↓ | 0.01↓ | 1.11↓ |
| Con | spring | 6.9 | 3.15↓ | 0.29↓ | 0.02↓ | 0.60↓ | 0.01↓ | 1.90↓ |
| | summer | 6.4 | 3.12↓ | 0.32↓ | 0.01↓ | 0.86↓ | 0.01↓ | 3.30↓ |
| | autumn | 7.3 | 2.28↓ | 0.31↓ | 0.03↓ | 0.83↓ | 0.01↓ | 2.50↓ |
| Cer | spring | 6,4 | 2.50↓ | 0.29↓ | 0.02↓ | 0.60↓ | 0.01↓ | 1.90↓ |
| | summer | 6.3 | 1.50↓ | 0.28↓ | 0.03↓ | 0.52↓ | 0.01↓ | 1.45↓ |
| | autumn | 7.2 | 1.40↓ | 0.28↓ | 0.01↓ | 0.94↓ | 0.01↓ | 2.10↓ |
| Epo | spring | 6.4 | 4.70↓ | 0.21↓ | 0.02↓ | 0.76↓ | 0.01↓ | 1.91↓ |
| | summer | 5.9 | 4.02↓ | 0.20↓ | 0.01↓ | 0.93↓ | 0.01↓ | 5.12↓ |
| | autumn | 6.6 | 3.10↓ | 0.30↓ | 0.04↓ | 0.97↓ | 0.01↓ | 3.30↓ |
| Gs | spring | 6.2 | 1.15↓ | 0.47↓ | 0.01↓ | 0.29↓ | 0.01↓ | 1.70↓ |
| | summer | 6.1 | 1.80↓ | 0.15↓ | 0.02↓ | 0.60↓ | 0.01↓ | 1.32↓ |
| | autumn | 6.2 | 1.03↓ | 0.09↓ | 0.02↓ | 0.08↓ | 0.01↓ | 2.15↓ |

*2.4. Assessment of Rainwater Microbiological Quality*

The assessment of changes in microbiological quality was based on:

- Determination of the number of bacteria *Escherichia coli* (*E. coli*), *Enterococci*,
- Determination of the number of bacteria *Clostridium perfringens*,
- Determination of the total number of bacteria at 20 and 37 °C,
- Luminometric determination of adenosine triphosphate (ATP) concentration.

All procedures were described as provided below, respectively.

2.4.1. Determination of the Number of Bacteria *Escherichia coli*, *Enterococci*

Membrane filtration in accordance with EN ISO 9308-1: 2004 and EN ISO 7889-2: 2004 was used to determine the number of bacteria *Escherichia coli* and *Enterococci*. Filtration was performed on a filter set (Merck, Darmstadt, Germany), using cellulose membrane filters (Merck Millipore, Darmstadt, Germany) with a total diameter of 0.45 mm and a pore diameter of 0.2 μm. Each time 100 mL of the sample was filtered and the filter was transferred to a medium: Chromocult (Argenta, Statens Serum Institute, Copenhagen, Denmark) for the *E. coli* assays and SLANETZ and BARTLEY agar (Merck, Darmstadt, Germany) for *Enterococci* detection. Each time, incubation was carried out at 37 °C for 48 h. The result was expressed in colony-forming units in 100 mL of water (CFU/100 mL).

2.4.2. Determination of the Number of Bacteria *Clostridium perfringens*

*Clostridium perfringens* spore bacilli were determined in accordance with EN ISO 14189: 2016-10. The Wilson–Blair II medium was used, supplemented with 8% iron (III) chloride (10 mL/L), 10% sodium hydroxide (6 mL/L) and 20% sodium sulfate (100 mL/L). The cultivation was carried out under anaerobic conditions at 37 °C. Characteristic dark colonies were verified as a positive. The result was expressed in colony-forming units in 100 mL of water (CFU/100 mL).

2.4.3. Determination of the Total Number of Bacteria at 20 and 37 °C

The total number of bacteria were determined using Koch's plate method with the use of culture-based methods in accordance with the Polish standard EN ISO 6222: 2004. R2A (Oxoid, Hampshire, UK) agar medium was used for the culture. Incubation for the total number of bacteria at 20 °C (psychrophilic) was carried out for 72 h at 20 °C, and for the total number of bacteria at 37 °C (mezophile) for 48 h. The results were expressed in colony-forming units in 1 mL of tested water (CFU/mL).

#### 2.4.4. Luminometric Determination of ATP Concentration

ATP evaluation was performed using a luminometer (LiminUltra, Fredericton, JCT, Canada) and a set of BacTiter reagents—Glo Microbial Cell Variability Assay (PROMEGA, Madison, WI, US). The analyses included the amounts of ATP from potentially living cells. For this purpose, the content of total ATP and extracellular ATP was determined (the assay protocol was based on the Promega Corporation guidelines) (Figure 2). The difference between these two measurements determines the amount of intracellular ATP in potentially living cells (general ATP—external ATP = internal ATP). Tested samples and the reagents for each run were kept at a constant temperature of 37 °C. Determination of the amount of extracellular ATP was preceded by filtration of the sample through sterile 0.2 μm syringe filters. The results were expressed in relative light units in 100 μL of water (RLU/100 μL).

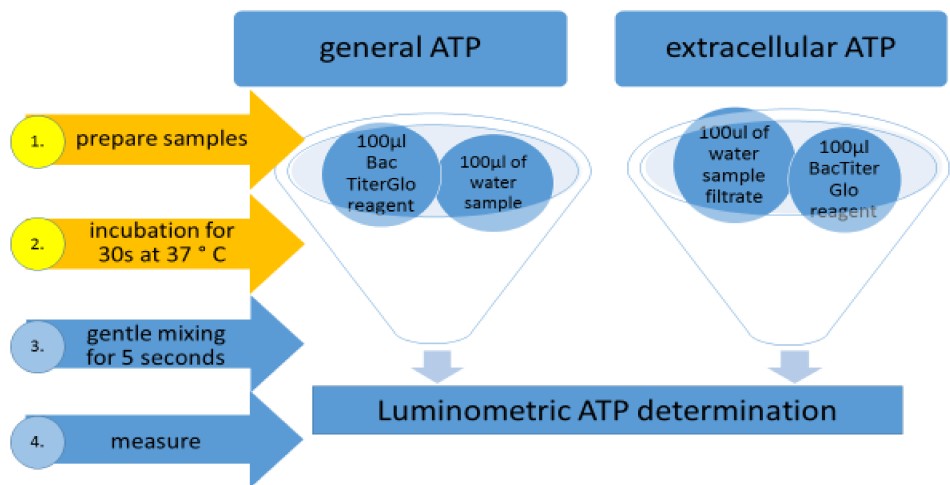

**Figure 2.** Procedure of sample preparation for luminometric determination of ATP concentration.

#### 2.5. Statistical Analysis—Collection and Interpretation of Data

The analysis of the correlation between the microbiological and physicochemical parameters was performed using Pearson dependence.

### 3. Results

#### 3.1. Determination of Microbiological Changes of Rainwater during Storage Time

Besides *Escherichia coli*, indicators such as *Enterococci, Clostridium perfringens* and the total number of bacteria at 20 °C and 37 °C were used to assess the risk of bacteriological contamination of water.

The concentration values for *Escherichia coli, Enterococci, Clostridium perfringens* indicator determined during the storage time, against collections from different roof surfaces and seasonal sampling, are presented in Tables 4–6 provided below.

#### 3.1.1. *Escherichia coli*

The results indicated that *E. coli* obtained a concentration of 50 CFU/100 mL, which is the highest number detected in the water collected from a flat roof covered with epoxy resin (Table 4). However, samples collected from other roof surfaces did not exceed the limit value for water for drinking purposes intended for consumption in crisis conditions (10 CFU/100 mL).

It was noted that the number of bacteria in the collected rainwater depends on the seasonal sampling and storage duration. Furthermore, in spring, only individual *E. coli* were detected in water collected from a galvanized sheet roof, whereas in the summer, *E. coli* presence was found in all stored rainwaters and they were the most numerous in comparison to spring or autumn.

**Table 4.** The number of *E. coli* in rainwater samples depending on storage duration, roof surface and seasonal sampling.

| Day of Storage | Spring | | | | | Summer | | | | | Autumn | | | | |
|---|---|---|---|---|---|---|---|---|---|---|---|---|---|---|---|
| | P | Con | Cer | Epo | Gs | P | Con | Cer | Epo | Gs | P | Con | Cer | Epo | Gs |
| | Number of *Escherichia coli* [cfu/100 mL] | | | | | | | | | | | | | | |
| 1 | 0 | 0 | 0 | 0 | 5 | 4 | 3 | 2 | 40 | 1 | 1 | 2 | 2 | 17 | 2 |
| 14 | 0 | 0 | 0 | 0 | 1 | 0 | 2 | 2 | 50 | 2 | 0 | 0 | 0 | 12 | 0 |
| 28 | 0 | 0 | 0 | 0 | 0 | 0 | 6 | 9 | 50 | 2 | 0 | 0 | 0 | 1 | 0 |
| 42 | 0 | 0 | 0 | 0 | 0 | 0 | 0 | 0 | 0 | 0 | 0 | 0 | 0 | 1 | 0 |
| 56 | 0 | 0 | 0 | 0 | 0 | 0 | 0 | 0 | 0 | 0 | 0 | 0 | 0 | 0 | 0 |
| 70 | 0 | 0 | 0 | 0 | 0 | 0 | 0 | 0 | 0 | 0 | 0 | 0 | 0 | 0 | 0 |

The study showed that with increasing storage duration, the microbiological quality of water improved due to decreasing number of occurring *E. coli*.

*E. coli* was detected for the longest storage duration (six weeks of storage) in rainwater collected from a flat roof covered with epoxy resin in autumn whereas the largest number of such bacteria were detected on 28 days of storage in the summer in rainwater collected from the same epoxy-roofing (50 CFU/100 mL).

### 3.1.2. *Enterococci*

Namely, the highest number of bacteria was detected in the summer sampled rainwater collected from roof covered with epoxy resin and ceramic tiles (Table 5). The obtained results indicate that even several dozen CFU/100 mL of *Enterococci* were detected in rainwater until the 28th day of storage duration.

**Table 5.** The number of *Enterococci* in rainwater samples depending on storage duration, roof surface and seasonal sampling.

| Day of Storage | Spring | | | | | Summer | | | | | Autumn | | | | |
|---|---|---|---|---|---|---|---|---|---|---|---|---|---|---|---|
| | P | Con | Cer | Epo | Gs | P | Con | Cer | Epo | Gs | P | Con | Cer | Epo | Gs |
| | Number of *Enterococci* [cfu/100 mL] | | | | | | | | | | | | | | |
| 1 | 0 | 1 | 1 | 4 | 4 | 7 | 2 | 20 | 57 | 0 | 0 | 2 | 1 | 37 | 0 |
| 14 | 0 | 0 | 0 | 3 | 0 | 0 | 5 | 55 | 60 | 0 | 0 | 0 | 0 | 7 | 0 |
| 28 | 0 | 0 | 0 | 0 | 0 | 0 | 2 | 50 | 77 | 0 | 0 | 0 | 0 | 0 | 0 |
| 42 | 0 | 0 | 0 | 0 | 0 | 0 | 0 | 0 | 0 | 0 | 0 | 0 | 0 | 0 | 0 |
| 56 | 0 | 0 | 0 | 0 | 0 | 0 | 0 | 0 | 0 | 0 | 0 | 0 | 0 | 0 | 0 |
| 70 | 0 | 0 | 0 | 0 | 0 | 0 | 0 | 0 | 0 | 0 | 0 | 0 | 0 | 0 | 0 |

### 3.1.3. *Clostridium perfringens*

Among the samples tested for the presence of *Clostridium perfringens*, spring sampled rainwater harvested from a roof covered with concrete tiles, ceramic tiles and a flat roof covered with epoxy resin were positive for *Clostridium perfringens*. Their presence remained in the stored water for about 14 days (Table 6).

### 3.1.4. The Total Number of Bacteria at 20 °C

A high number of total number of bacteria at 20 °C was detected in summer sampled rainwater ranging up to $7 \times 10^3$ CFU/mL and spring sampled rainwater ranging up to 3500 CFU/mL. The results indicated that the lowest number of bacteria was evaluated for rainwater collected from a roof covered with a galvanized sheet. Furthermore, regardless of the season and roof catchment surface, the total number of bacteria at 20 °C of stored water decreased (Figures 3–5).

### 3.1.5. The Total Number of Bacteria at 37 °C

According to the number of bacteria at 37 °C, the highest number was recorded for rainwater collected in summer from a flat roof covered with epoxy resin ranging up to $6 \times 10^3$ CFU/mL. However, the lowest concentration of these bacteria was ranging only a few dozen CFU/mL for the samples collected in spring and summer from the roof covered with galvanized sheet metal. Autumn sampled water indicated these bacteria only in water harvested from the resin-covered roof. Furthermore, with regard to the extension of the storage time duration, decreasing tendency for the number of bacteria to at 37 °C was noticed (Figures 6–8).

**Table 6.** The number of *Clostridium perfringens* in rainwater samples depending on storage duration, roof surface and seasonal sampling.

| Day of Storage | Spring | | | | | Summer | | | | | Autumn | | | | |
|---|---|---|---|---|---|---|---|---|---|---|---|---|---|---|---|
| | P | Con | Cer | Epo | Gs | P | Con | Cer | Epo | Gs | P | Con | Cer | Epo | Gs |
| | Number of *Clostridium perfringens* [cfu/100 mL] | | | | | | | | | | | | | | |
| 1 | 0 | 2 | 1 | 3 | 0 | 0 | 0 | 0 | 0 | 0 | 0 | 0 | 0 | 0 | 0 |
| 14 | 0 | 0 | 0 | 1 | 0 | 0 | 0 | 0 | 0 | 0 | 0 | 0 | 0 | 0 | 0 |
| 28 | 0 | 0 | 0 | 0 | 0 | 0 | 0 | 0 | 0 | 0 | 0 | 0 | 0 | 0 | 0 |
| 42 | 0 | 0 | 0 | 0 | 0 | 0 | 0 | 0 | 0 | 0 | 0 | 0 | 0 | 0 | 0 |
| 56 | 0 | 0 | 0 | 0 | 0 | 0 | 0 | 0 | 0 | 0 | 0 | 0 | 0 | 0 | 0 |

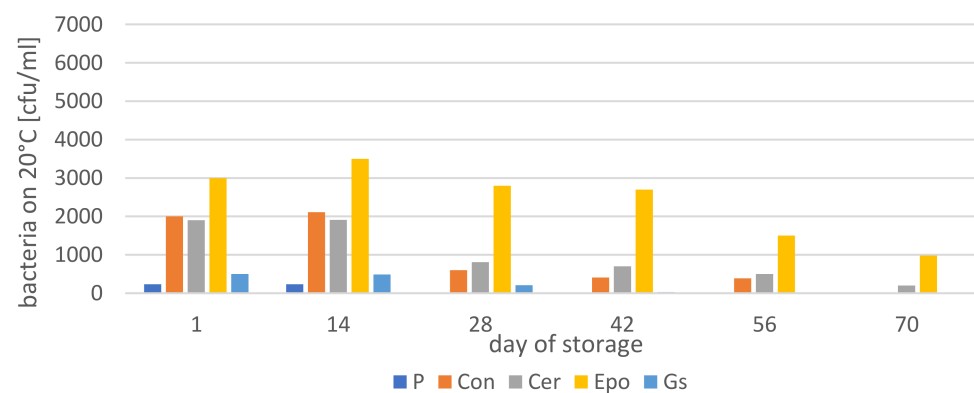

**Figure 3.** The total number of bacteria at 20 °C in spring sampled rainwater depending on storage duration, roof surface.

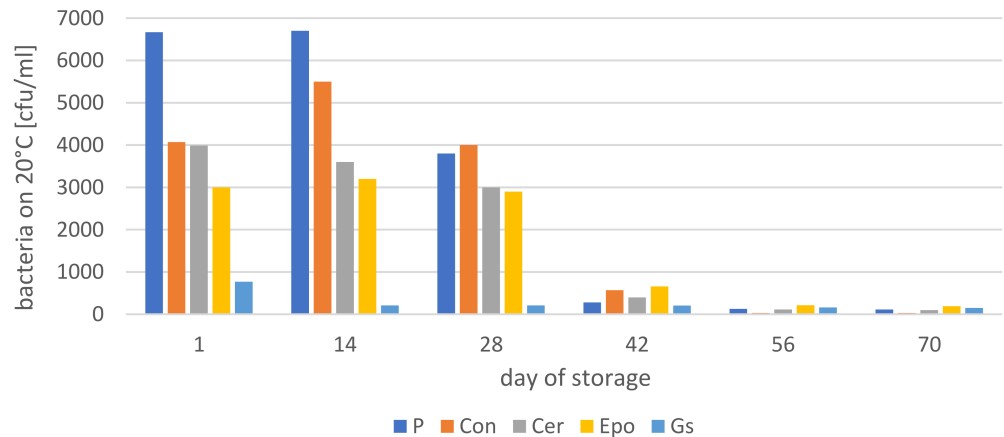

**Figure 4.** The total number of bacteria at 20 °C in summer sampled rainwater depending on storage duration, roof surface.

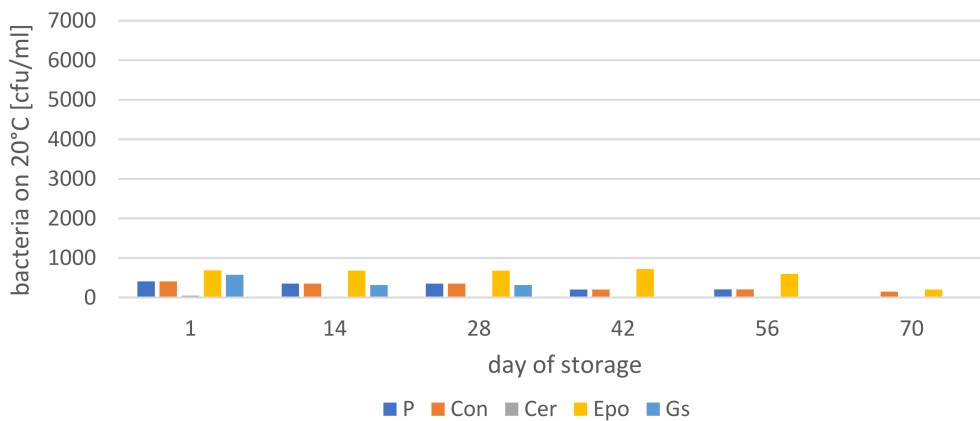

**Figure 5.** The total number of bacteria at 20 °C in autumn sampled rainwater depending on storage duration, roof surface.

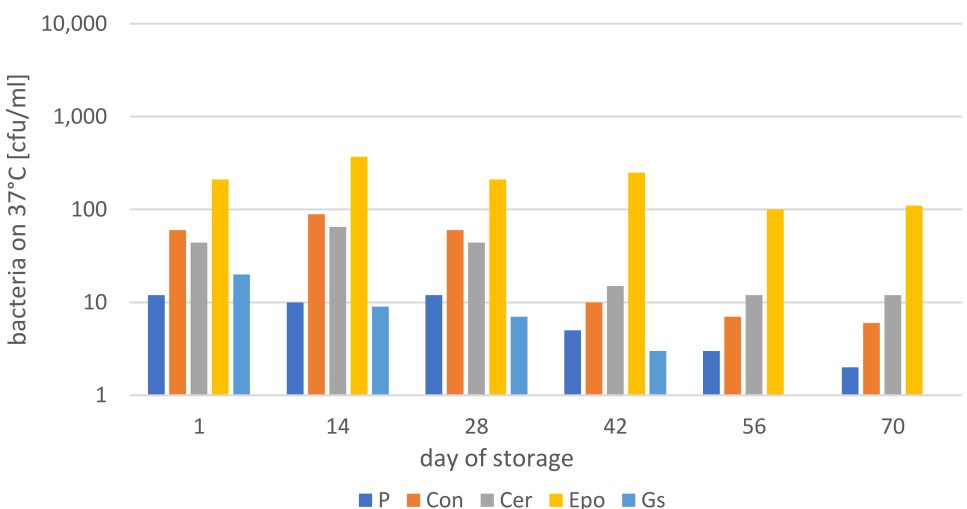

**Figure 6.** The total number of bacteria at 37 °C in spring sampled rainwater depending on storage duration, roof surface.

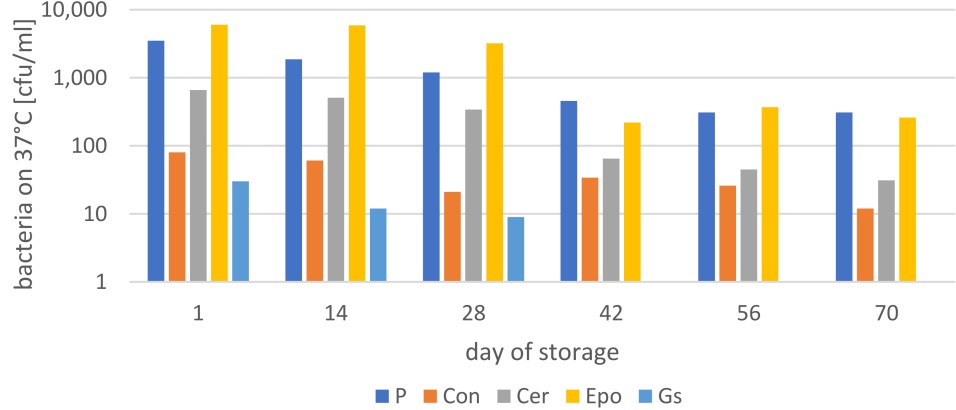

**Figure 7.** The total number of bacteria at 37 °C in summer sampled rainwater depending on storage duration, roof surface.

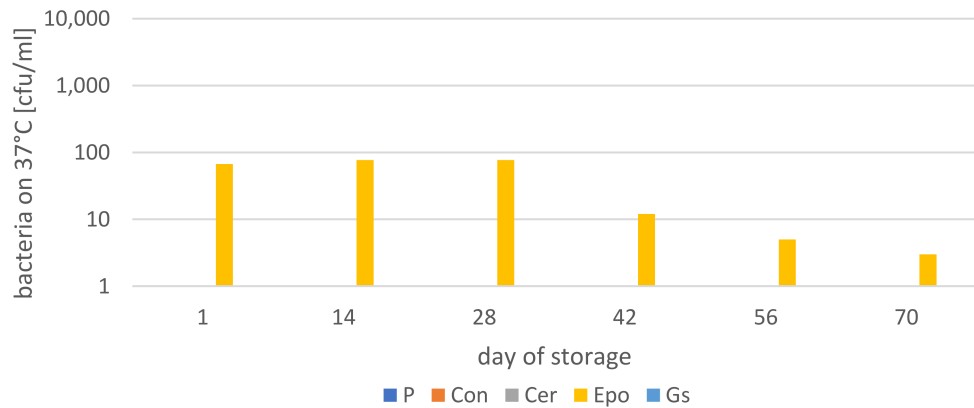

**Figure 8.** The total number of bacteria at 37 °C in autumn sampled rainwater depending on storage duration, roof surface.

### 3.1.6. ATP Concentration

The results indicated that the highest ATP concentration was detected in the water collected from a flat roof covered with epoxy resin in the spring (range up to 28,000 RLU/100 μL) and summer (range up to 20,000 RLU/100 μL).

Moreover, as in the case of the total number of bacteria, the highest ATP concentration was found for rainwater collected from a flat roof covered with epoxy resin, and the lowest for rainwater collected from a roof covered with galvanized steel sheet. Furthermore, for longer storage durations, a decreasing ATP concentration was observed (Figures 9–11).

Concerning the microbial quality of stored rainwater, it was assessed that initial rainwater quality compared to the rate of bacterial reduction form a direct relationship dependency on the roof covering type:

$$Gs > Cer > Con > Epo.$$

Additionally, the lowest number of bacteria was found in rainwater collected from the roof covered with galvanized sheet metal. Moreover, it was also found that samples of rainwater collected in autumn performed the best in terms of microbiological quality and the following dependency was evaluated:

$$autumn > spring > summer.$$

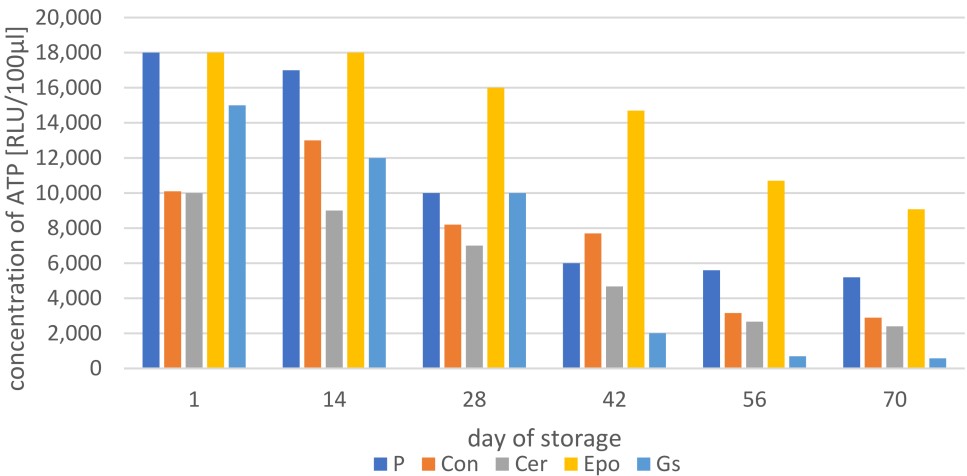

**Figure 9.** Concentrations of ATP for spring sampled rainwater depending on storage duration, roof surface.

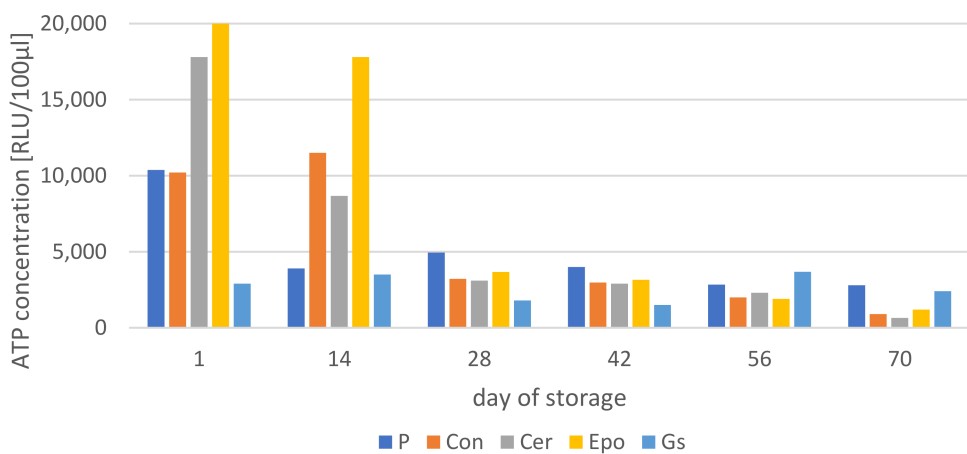

**Figure 10.** Concentrations of ATP for summer sampled rainwater depending on storage duration, roof surface.

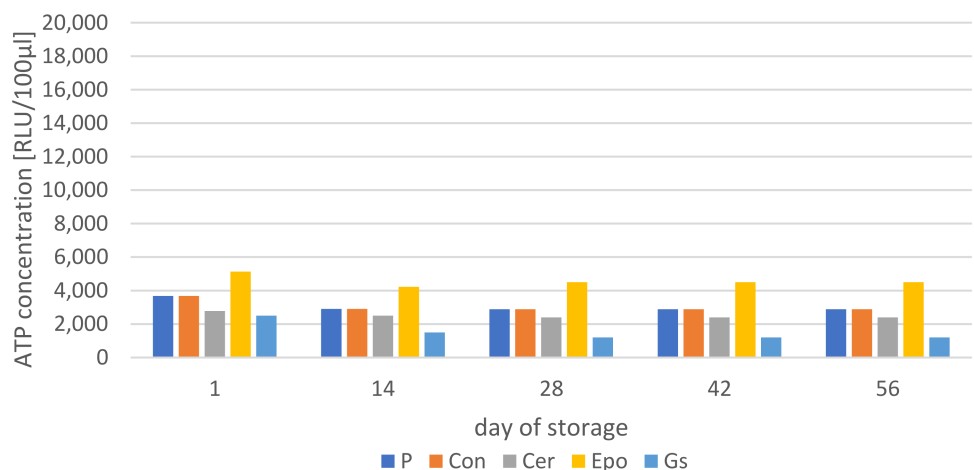

**Figure 11.** Concentrations of ATP for autumn sampled rainwater depending on storage duration, roof surface.

Finally, based on the collected data, authors concluded that variations in the microbiological quality of tested rainwater during its storage indicate an improvement in the sanitary safety in terms of fecal contamination indicator microorganisms. However, the total number of bacteria does not meet the requirements for drinking water. Hence, the disinfection process is recommended before potable use.

*3.2. The ATP Correlation to Indicator Organisms*

Various positive correlations were observed between the concentrations of ATP and the indicator organisms detected in the studied samples, namely, the total number of bacteria 20 °C and 37 °C. Furthermore, based on the correlations and concurrence frequencies observed, the authors highlight benefits for ATP concentration analysis in case of rapid assessment of microbial rainwater quality. Moreover, positive correlations between the total number of bacteria 20 °C and 37 °C and seasonal sampling and roof surface type were provided (Table 7).

The correlation study showed that for rainwater harvested and stored in spring, at least high correlations were found between the total number of bacteria and the ATP content. In summer and autumn, these correlations values were mostly high (for rainwater collected from a roof covered with a galvanized sheet). Therefore, a strong dependency between the total number of bacteria and the ATP concentration was observed.

**Table 7.** Correlation coefficients between ATP concentration and the total number of bacteria in rainwater samples depending on roof surface type.

| Rainwater Collected From: | Spring | | Summer | | Autumn | |
|---|---|---|---|---|---|---|
| | **Mesophilic** | **Psychrophilic** | **Mesophilic** | **Psychrophilic** | **Mesophilic** | **Psychrophilic** |
| precipitation (directly) | 0.827851 | 0.953434 | 0.918316 | 0.674004 | - | 0.627229 |
| roof covered with concrete tiles | 0.897749 | 0.873979 | 0.645955 | 0.384902 | - | 0.627229 |
| roof covered with ceramic tiles | 0.895669 | 0.945296 | 0.795021 | 0.665168 | - | 0.924684 |
| epoxy resin roof | 0.94229 | 0.883441 | 0.68997 | 0.410188 | −0.5902 | −0.22933 |
| roof covered with galvanized steel sheet | 0.916805 | 0.959076 | 0.19068 | 0.112606 | - | 0.827761 |

**0.9 ≤ r < 1.0**—almost complete correlation, **0.7 ≤ r < 0.9**—very high correlation, **0.5 ≤ r < 0.7**—high correlation.

## 4. Discussion

The harvesting, storing and using of rainwater for drinking and hygienic purposes in crisis conditions discussed in this article are crucial for the sanitary safety and human health risk of rainwater use. However, the results of the microbiological analysis of the stored rainwater clearly indicate the presence of pathogenic indicator microorganisms, which depending on the conditions can multiply and be dangerous for users [30,31]

In the literature, there is a lack of information available on monitoring changes in rainwater quality during the storage process. For this reason, it is difficult to confront the obtained results with the conclusions of other researchers. Among published studies, most usually concern the comparison of the microbiological quality of the first runoff water with the water intended for storage and indicate a better quality of water from tanks, regardless of the roofing from which rainwater was collected [32–35].

For physicochemical and microbiological quality changes, it was noted that all values of controlled parameters decreased (Tables 2 and 3), except for the pH, which fluctuated in the range of 6.0–7.3 depending on the sampling season and type of roofing material. Despins states that changes in the pH value of stored rainwater highly depend on the material from which the tank is made. The pH value usually increases with the precipitation intensity and it was higher in plastic tanks than in the concrete tanks [36].

This was confirmed for rainwater with pH 5.7, which, after storage in concrete tanks, increased to 6.7 and after storage in plastic tanks, up to 8.7 [37]. In metal tanks, the pH was significantly lower, on the contrary to tanks made of cement-based materials [38,39]. The results obtained in this study do not confirm the mentioned data. In the presented studies, no significant correlations were found between changes in the number of bacteria and changes in the pH of tested samples.

This study showed that the storage of rainwater improves its quality, mainly due to the process of sedimentation of particles at the bottom of the tank and consequently determine the turbidity of the water. The highest initial turbidity 7.7 NTU was obtained for samples collected from the roof covered with epoxy resin. Around the 20th day of storage, the turbidity value decreased to 2 NTU and remained at this level for all stored waters. A similar dependence was demonstrated for freshly harvested rainwater. The turbidity was determined at the level of 3.0–6.5 NTU, and after 4 weeks of storage, 2.0–3.5 NTU [39,40].

The qualitative microbial assessment of rainwater required the determination of *Escherichia coli* and the total number of bacteria [30,31]. Furthermore, the presence of *Escherichia coli* in drinking water indicates fecal contamination and also may contain other harmful or pathogenic organisms, including bacteria, viruses or parasites. Thus, detection of *E. coli* in water in the range of 1–3 cfu/mL; 4–6 cfu/mL and 7–9 cfu/mL was considered as low, medium and high human health risk, respectively [41].

The use of storage as a method for improving the sanitary quality of rainwater is possible in the case with closed tanks (no fresh rainwater inflow) and a sufficiently long retention time. This study verified that to improve microbiological parameters, a storage period not shorter than six weeks, was required. If it is necessary to use rainwater as a

source of drinking water in advance, disinfection is necessary to ensure water sanitary safety and under certain conditions biological stability. Such conditions were fulfilled in the studies performed by us, in which observations concern static conditions using relatively small water volumes (30 L) and without fresh sprouts of rainwater. Thus, biostable water does not contain microorganisms and does not support their development in the storage tank. The instability of water, resulting in the development of biofilm on the internal surfaces of the storage tank and installation pipes, low organoleptic properties of water and secondary contamination with pathogenic bacteria [42].

There is limited information available regarding biofilm formation and development in water distribution systems [43]. However, there is no doubt that the presence of nutrients (inorganic nitrogen and phosphorus compounds and the presence of dissolved biodegradable organic substances) in water is the main factor behind microorganism development, which results in unstable water [44–46]. For example, The threshold values of parameters limiting secondary development of microorganisms in distribution systems should be lower than 0.2 gNnorg/m$^3$ 0.03 gPO$_4^{3-}$/m$^3$ and: 0.25 gC/m$^3$ BDOC assuming that the BDOC content is about 10.6% DOC [47].

The determined nutrient concentration in the stored rainwater samples indicates a sufficient amount of presence of nutrients for microorganisms development (Table 3). The results obtained after 28 days of storage (Table 4) show a reduced amount of nutrients in relation to the parameters of rainwater on the first day of storage. Therefore, improved microbiological quality of rainwater directly depends on decreasing nutrients concentration in the tested samples.

For waters containing TOC and inorganic nitrogen, phosphate ions are most relevant [48,49]. An insufficient amount of phosphate ions inhibits microorganism development more than in the case of other nutrients [45]. It should also be noted that phosphorus as well as other nutrients in the first days of use of storage systems, as well as water installations made of plastic, may be released from these materials, causing faster microorganisms development [50]. It seems, however, that the phosphorus research carried out by the authors was a limiting factor and its deficiencies have reduced the number of microorganisms during rainwater storage.

The number of microorganisms is certainly also influenced by sedimentation. This is how natural sedimentation processes are used to produce clearer water with fewer microorganisms. As described in the introduction, this will give an indication of the degree of contamination of the water before and after storing it for several weeks. Thus, it will be possible to estimate the costs connected with its final treatment for the needs of various branches of the economy and for drinking in crisis conditions.

Temperature is another essential parameter determining the microbiological water quality. Moreover, temperature determines the critical environmental factors in creating the taxonomic composition of rainwater bacteria. Although, there are limited data on the composition of bacteria in rainwater stored at different temperatures, dynamics of bacterial aggregations in rainwater at different seasons and temperatures [32,51]. The different responses of microorganisms to temperature indicate the need for comprehensive rainwater management practices at different seasons. For example, additional disinfection may be needed to eradicate bacteria in warm seasons or in tanks exposed to sunlight.

It was observed that during the rainwater storage process, various microorganisms reflect specific temperature reactions, for example, longer persistence of *E. coli* was detected in rainwater at 4 °C, which is consistent with the data on the survival of pathogenic microorganisms [52].

It was noted that in rainwater samples collected from a roof covered with epoxy resin, E. coli bacteria were detected even after more than 40 days of storage at 12 °C. However, the elimination rate of many pathogenic bacteria at 4 °C is much slower compared to the temperature at 20 °C and 37 °C. The maximum survival time of bacteria is approximately two times longer at lower temperatures and is about 140 days, which suggests the same conclusions as described above [53].

Furthermore, the presented study seems to be justified also due to often implemented closed underground storage solutions according to the use of rainwater, for example, at newly designed estates of single-family houses (used for aesthetic and visual reasons and also due to limited space) [51,52]. The storage temperature of rainwater in underground containers is constant. Hence, in this case, it was assumed that the season in which rainwater was collected plays a more important role [54,55]. This study demonstrates that namely the number of *E. coli* exceeded the limited level for both summer and autumn, and rainwater collected and stored in the summer season can be safely used only after six weeks of storage (Table 4). The maximum number of *E. coli* in the rainwater collected from the epoxy resin roof was 50 CFU/100 mL.

Fecal *Enterococci* are included in the European Union Drinking Water Directive as one of the two basic parameters describing the sanitary quality of water intended for safe human use. The presence of these microorganisms in drinking water is not acceptable, and their detection indicates contamination that may threaten human health. Their presence is noted not only in rainwater or surface water but also, according to some studies, in water subjected to chemical disinfection with chlorine preparations [56]. Therefore, their presence in rainwater and long-term survival in storage tanks (even above 50 jtk/100 mL on day 28 of storage) should not be surprising.

The study also included the determination of the number of spore-forming bacteria—*Clostridium perfringens*. This is an important microbiological indicator, and the presence of these microorganisms in the water may indicate fecal contamination that occurred even months earlier. A characteristic feature of these organisms is their very long survival in aquatic environments. *Clostridium perfringens* can be detected even when no *E. coli* or fecal *Enterococci* are recorded. Its determination allows to fully describe the quality of tested water and indicate the safety of its use [57]. Referring to the studies performed, the results indicate the presence of *C. perfringens* only in rainwater collected during the spring season. As of day 28, they were no longer listed. The absence of these bacteria may indicate the sanitary safety of the stored rainwater.

On average, the occurrence of coliform bacteria was significantly higher when water temperatures were above 15 °C. Temperature is widely recognized as an important controlling factor in influencing bacterial growth. In climates where water temperatures are warm, bacterial growth may be very rapid. However, the minimum temperature at which microbial activity was observed varied from system to system. Systems that typically experienced cold water had increases in coliform occurrences when water temperatures ranged near 10 °C. The strains of coliform bacteria in these systems may be better adapted to grow at lower temperatures (psychrophiles). Therefore, it seems that the most appropriate storage of rainwater is at a temperature of about 10 °C. In subtropical or temperate climate zones, such temperatures can be maintained by collecting rainwater in underground tanks.

In the case of the number of heterotrophic bacteria determined at 22 ± 2 °C, mainly bacteria constituting the autochthonous microflora of water were detected, which are not of significant sanitary importance, as they do not have the ability to multiplicate at human body temperature (37 °C) and are therefore usually harmless to human health.

Among the bacteria isolated from water, which show growth at 22 ± 2 °C, the following may be mentioned: bacteria of the *Pseudomonas*, *Flavobacterium*, *Bacillus genera*, as well as strains of bacteria which are detected at higher temperatures and therefore potentially pathogenic (e.g., bacteria of the *Vibrio*, *Serratia*, *Proteus*, *Staphylococcus* and *E. coli* genera). These microorganisms generally do not pose a threat to human health, but some of them may be opportunistic pathogens and pose a risk to people with immunodeficiency of various origins. According to the directive of the European Union on the quality of drinking water, the number of bacteria from this group must not exceed 100/mL. In the studied rainwater, in the first few weeks of storage, an excessive number of bacteria mentioned above was recorded [58].

Most research studies on assessing the microbiological quality of water report that culture-based methods are not accountable factors and are inaccurate in terms of the num-

ber of microorganisms determined. The reliability of such methods as a health-based indicator for monitoring rainwater quality has been questioned in the literature. Most of the studies in the literature show that only a small percent of common microorganisms present in the natural environment are able to proliferate in the same way under laboratory conditions [59]. The studies in the literature state that the determination of ATP concentration is a valuable method for the measurement of the variability of biological contamination. Besides this, the main advantage of this method is the short testing time and how results are obtained straight away. Additionally, there are luminometers that allow for ATP determination in situ (on the spot), without the need to transport the samples to the laboratory space [60,61]. However, originally, the luminescent ATP method was used to check the level of microbiological purity on solid surfaces (food industry, food packaging production, etc. [62]. Over the last years, it has also been tested to determine the level of microbial contamination in water [50,63].

Based on our study, it was found that ATP determination can be used for monitoring changes in the quality of stored rainwater, due to high correlations of ATP changes with the total number of bacteria at 37 °C and 20 °C. This is also confirmed by studies published by other researchers [64,65]. However, it is impossible to indicate in which range of ATP concentration water can be considered as sanitary safe.

Even low ATP concentrations in water may result from the presence of pathogenic microorganisms. What is more, it is important that the ATP determination is not a selective method and determines the total ATP in the sample. Water samples may contain a significant amount of extracellular ATP, which is not equal to a sanitary health risk as the sample may contain non-proliferating dead microorganisms [66].

The obtained results indicate that rainwater can be considered safe only after disinfection. However, there are limitations to the use of ATP determination to control the effectiveness of disinfection (in particular for chlorine and its compounds disinfection). This is not possible due to the biochemical reactions between the reagents used in the determination of ATP (luciferin) and $Cl^-$ ions [67].

The conducted research seems reasonable also due to the increasingly common solutions related to the use of rainwater in the newly-reign single-family housing estates [68,69]. There is usually proposed for rainwater storage in closed underground tanks mainly due to limited space and aesthetic considerations. To safely use these resources, especially for purposes requiring the best quality waters, the potentially most advantageous conditions for collecting rainwater and storage should be pointed out. It seems important to designate parameters by means of which the fastest and most equivalent to the quality of microbiological rainwater and the possibility of their safe use. Based on the resulting results over the quality of rainwater and their changes during storage, the following directions of further research were determined:

- Checking changes in the quality of microbiological rainwater in tanks in real conditions at a fixed recovery of fresh rainfall,
- Further research on the level of parameters limiting the development of microorganisms in stored rainwater,
- Checking the effectiveness of various rainwater disinfection methods to obtain sanitary safety of its users—conducting statistically significant research on the ability to use ATP as a quick and direct method of assessing the quality of microbiological rainwater.

## 5. Conclusions

The findings of the current work conclude that:

- The results obtained in this study show that rainwater retention improves its physico-chemical and microbiological quality, and does not depend on the season and the type of roofing material from which it was collected.
- The use of storage as a method for enhancing the sanitary quality of rainwater is possible in the case of closed tanks (with no fresh rainwater inflow) and sufficient retention time. This study verified that to improve the microbiological parameters of

- stored rainwater, a storage period not shorter than six weeks, was required. If it is necessary to use rainwater as a source of drinking water in advance, disinfection is required to ensure biological stability.
- Regarding the microbial quality of stored rainwater, it was assessed that initial rainwater quality compared to the rate of bacterial reduction forms a direct relationship dependency on the roof covering type: Gs > Cer > Con > Epo.
- Furthermore, the lowest number of bacteria was found in rainwater collected from the roof covered with galvanized sheet metal. Moreover, it was also found that samples of rainwater collected in autumn performed the best microbiological quality and the following dependency was evaluated: autumn > spring > summer.
- The worst microbiological quality was characterized by rainwater collected in spring and autumn with a flat roof covered with epoxy resin.
- However, at the current stage of this research, it is not possible to indicate in which range of ATP concentration water can be considered as sanitary safe (low ATP concentration does not exclude the presence of pathogenic microorganisms).
- Due to the strong dependency between the total number of bacteria (20 °C and 37 °C) and the ATP concentration, the potential use of the fast luminometric analytical method for monitoring the microbiological quality of water was recommended.

**Author Contributions:** Conceptualization, M.Z., J.Z., D.P.; data curation, M.Z.; Investigation M.Z., J.Z., D.P.; methodology: M.Z., J.Z., D.P.; software: M.Z., A.S.-W.; validation, M.Z., J.Z., D.P., A.S.-W.; formal analysis: M.Z., D.P.; investigation: M.Z., J.Z., D.P.; data curation: M.Z., D.P.; writing—original draft preparation: M.Z., J.Z., D.P.; writing—review and editing: M.Z., D.P., A.S.-W.; visualization, M.Z., D.P., A.S.-W. All authors have read and agreed to the published version of the manuscript.

**Funding:** This research received no external funding.

**Institutional Review Board Statement:** Not applicable.

**Informed Consent Statement:** Not applicable.

**Data Availability Statement:** Not applicable.

**Conflicts of Interest:** The authors declare no conflict of interest.

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
