# Peer review of "Investigation of Microbiological Quality Changes of Roof-Harvested Rainwater Stored in the Tanks"

_resources, doi:10.3390/resources10100103_

Round 1

Reviewer 1 Report

General comment

The manuscript entitled “Investigation of Microbiological Quality Changes of Roof-Harvested Rainwater Stored in the Tanks” by Zdeb et al. investigated the physicochemical and microbiological quality of stored rainwater. The authors reported the optimal conditions for rainwater storage and the methodology for determining microbiological quality of stored rainwater. This study is interesting and provides important knowledges to utilize rainwater for drinking safely. Although this manuscript is within the scope of this journal, I think that the manuscript still has some problems. Therefore, I would suggest a major revision of the present manuscript before it can be accepted for publication.

Major comments

  • The authors collected the rainwater directly from atmospheric precipitation, and its microbial quality was shown. However, the microbial quality of the rainwater before storage (Day 0) was not included (Table 5–7, Figure3–11). I think that these data are important to evaluate the temporal changes in microbial abundances. For example, it can be hypothesized that many microorganisms attached to the roof covered with epoxy resin compared with other materials, which resulted in the increase of microbial abundance at day 1.
  • The authors determined the total number of bacteria at 20°C and 37°C, while the other microbes, E.coli, Enterococci, and C. perfringens, were cultivated at 37°C. Please add the objective of a total number of bacteria at 20°C. In addition, please add the discussion which cultivating temperature is suitable for determining the microbial quality of the rainwater.
  • The limit value for water for drinking purpose was described only for E.coli, not for Enterococci and C. perfringens. It is important whether did the number of these microbes exceed the limit.
  • As you mentioned in the Discussion, ATP is a not selective method and indicates the total number of organisms in the rainwater, not specific to the pathogenic microorganisms. In addition, the results of pathogenic microbes (Table 5–7) seem to be inconsistent with them of ATP concentrations (Fig 9–11). Therefore, I think that the results in the present manuscript can not recommend the potential use of luminometric analytical method for monitoring microbiological quality. I suggest the correlations between ATP concentration and indicator microorganisms (E.coli, Enterococci, and C.perfringens) to be included in Table 8 if the significant correlations were observed.

Minor comments

  • L30: The authors should present the full form of RWHS. (The first time you use this word in the main manuscript)
  • L39: What is the full form of WFD?
  • L59: E.coli => Escherichia coli (The first time you use this in the main manuscript)
  • L130: tab.2 => Table 2
  • L143: tab.3 => Table 3
  • L157: Please confirm the unit of cellulose membrane filters (a total diameter of 0.45 mm).
  • L204: Table 4 is missing.
  • Please be consistent with the units you use (e.g. /L, /l, CFU, cfu, etc).
  • Table 8: indicator organisms => total number of bacteria

Reviewer 2 Report

Comments to the authors

A brief summary

The depletion of natural water resources is an up-to-date theme among the scientific society and developing of new methods for water collecting and storage is essential. In this regard the manuscript, titled “Investigation of Microbiological Quality Changes of Roof-Harvested Rainwater Stored in the Tanks”, that was given to me for review is very topical. The main purpose of the study is to investigate the microbiological quality of collected rainwater after long period of storage (3 months) in PEHD canisters. The water was collected during three seasons – spring, summer and autumn from roofs covered with: concrete tile, ceramic tile, galvanized sheet, epoxy resin and directly from the air (airborne microorganisms). The results obtained showed that during storage the main biogenic substances and the turbidity of the stored waters decrease as well as the amounts of live microorganisms. An alternative method for detection of live microorganisms in the samples was studied – measuring of the ATP concentration. The results showed almost complete correlation between ATP concentration and the number of psychrophilic microorganisms detected in almost all samples (except roof covered with concrete tiles) in spring.

I value the main idea of the study but I have some remarks and recommendation to authors which aim to increase the value of the article.

Introduction

Line 28: use “where” instead “were”;

Line 30: please, give the full name of this abbreviation. Rainwater harvesting systems I suppose... It is more appropriate the full name to be given here not at line 44;

Line 58: I suggest here the word "aforementioned" to be used instead “mentioned”. This suggestion is of a recommendatory nature;

Line 59: the name of the bacterium Escherichia coli must be given in full as it appears for the first time in the text here;

Lines 62-63: you wrote “at the stage of precipitation in the atmosphere”. Maybe you mean precipitation of airborne microorganisms? Am I right? If yes you should clarify this. Also, it is more correct to write "contaminated with microorganisms" not only "contaminated" on line 62.

Line 68: please, clarify what do you mean “precipitation”. Precipitation of what? It is not completely clear.

Line 77: I suggest here the word "Whereas" to be substituted with “however” in order the sentence to be clearer.  This suggestion is of a recommendatory nature;

Line 81: not very clear here. please rewrite this part of the sentence;

Line 84: “were” instead of “was” should be used;

 Materials and Methods

Section 2.2.

Line 116: what type of disinfection? It is good to specify this;

Polyethylene High-Density! Please, give the full name of this plastic canisters.

Line 126-128: you wrote “10 polyethylene cuvettes (40x40 cm) were placed 1.5 m above the ground level on a special stand construction. Finally, rainwater was collected in a 30 liters polyethylene tank.” Are these the control samples which contain only airborne microorganisms? If yes, it is good to be mentioned here.

Table 2: the full names of NTU and TOC should be given below the table. This suggestion is of a recommendatory nature;

Section 2.3.

Line 137-138: the sentence is not clear enough.

Line 140: “were” instead of “was” should be used;

Line 143: table 3 shows obtained results so my opinion is that this table should be moved to results section.

Page 6, table 3: I suppose that the arrows indicate the decrease of the tested parameters compared to the initial. I suggest here to notice that.

Other comments: nowhere in the text is mentioned the exact way the probes were taken for all analyses. The only involved information is:” Every 2 weeks samples was (the word must be substituted with were) taken for further analysis using peristaltic pump and sterile and disposable silicone tube.” Did you stir the water in the canisters prior to analyses? My opinion is that reliable results cannot be obtained without stirring the water in the tanks prior to analyses. This remark concerns all experiments you have done: microbiological and physicochemical. Please clarify the exact way of collecting the water samples from the tanks.

Section 2.4.

Line 154: the method is called "membrane filtration"! not “method of membrane filters”;

Line 159: for Enterococci detection”. Please add this information at the end of this sentence after the name of the media. The name of the medium is Slanetz and Bartley agar, without “a” at the end of the word Slanetza. Please, correct the name.

Line 160: I didn't see any controls. Did you use reference culture of E. coli and Enterococcus faecalis/faecium as positive controls in the experiments?

Line 164: the medium for Cl. perfringens detection is called Wilson-Blair, not Wilson-Braila. Please, correct the name of the medium.

Line 166: did you use reference culture of Cl. perfringens as positive control;

Results

Section 3.1.

Line 196: “were” instead of “are” should be used;

Line 212: in the Materials and methods section you wrote that the samples were taken every 2 weeks "using peristaltic pump and sterile and disposable silicone tube " My question is: did you homogenize the water in the canisters prior to collect the samples or not? My opinion is that a large number of microorganisms could precipitate at the bottom of the canister after 4/6 weeks of storage and collecting the water samples without homogenization is not the correct way to do it. Another acceptable option for obtaining reliable and complete results is to analyze the water from at least three point of each canister: top, middle and bottom.

Line 237: 7000 CFU/ml - it is not the accepted style of bacterial concentration expression. It must be given as 7x103 cfu/ml. This comment concerns all places in the text where bacterial concentration is indicated.

Section 3.2.

Table 8: The terms psychrophilic and mesophilic should be mentioned above in the Materials and Methods section. I suggest in section 2.4. Determination of the total number of bacteria at 20 and 37 ° C.

Concrete comments:

In order the results to become clearer I suggest to rewrite this section along with Materials and Methods section. Add the additional information, regarding the exact method of collecting the water from the canisters.

Discussion

Line 345: “This study showed that the storage of rainwater improves its quality, mainly due to the process of sedimentation of particles at the bottom of the tank and consequently determine the turbidity of the water. The highest initial turbidity 7.7 NTU was obtained for samples collected from the roof covered with epoxy resin. Around the 20th day of storage, the turbidity value decreased to 2 NTU and remained at this level for all stored waters.” - This comment sounds frivolously. The sedimentation of the particles is naturally occurring process in the tank which is stored static. But it is not appropriate to measure NTU without stirring the water in the tank prior to measure the turbidity.

Please, clarify somewhere in the paper maybe it is most appropriate in the Result section or Materials and methods section p.2.3., if you stirred the water in the canisters prior to analyses. Please, add some information in the text how you can explain water clarification during storage!

Line 367: “The results of obtained after 28 days of storage (Table 4) shows a reduced amount of nutrients in relation to the parameters of rainwater on the first day of storage. Therefore, improved microbiological quality of rainwater directly depend on decreasing nutrients concentration in the tested samples.” Comment: The relationship between microbial growth and number and the amount of the available nutrients in the water is absolutely clear. When nutrients are depleted, it stops the growth of microorganisms. Meanwhile, the number of the live microorganisms in the canisters increases. Not detecting live cells after 28 day of storage could be due to their precipitation at the bottom of the canisters.

Please, clarify here what is your interpretation of these results.

As it is clear from my comments above, my general remarks are linked with the methodology of the experiments and the results interpretation based on it. My general opinion is that the main idea of the study is good and should be supported but after revision.

Round 2

Reviewer 1 Report

The manuscript has been improved, and I resolved my misunderstanding. I recommend that the manuscript be accepted for publication after minor revision.

Minor comments

The authors should confirm whether there are mistakes in the manuscript thoroughly.

Line 128: Please add a space. [(HD-PE – 100)canisters => (HD-PE – 100) canisters]

Line 139: Please delete a space. [precipitations sampling “ => precipitations sampling”.]

Line 198: Please delete a comma after “The total number of bacteria”.

Line 205: There are two spaces after “37 °C (mezophile)”.

Table 6: There are two spaces after “The number of”.

Figure 6: Please revise the label of y axis. [on 37° => 37 °C]

Table 7: There are two spaces after “0,5≤ r < 0,7”.

Line 384: this studies => this study

Line 396: drinking water indicate => drinking water indicates

Line 435: sedimantation => sedimentation

Line 474: Please add a space before [56].

Line 526: ect. => etc.

Author Response

All comments of reviewers were taken into account and corrections were made in the text of the manuscript. In particular, part of the methodology has been supplemented with a detailed description of rainwater sampling from storage tanks. In addition, information on some microbiological indicators (faecal streptococci and Clostridium perfringens) was also provided, and their discussion was supplemented in the discussion. Following changes were made in the field of text editing and corrections from the English language, indicated by the reviewers.

Reviewer 2 Report

No additional comments to the authors.

Author Response

All comments of reviewers were taken into account and corrections were made in the text of the manuscript. In particular, part of the methodology has been supplemented with a detailed description of rainwater sampling from storage tanks. In addition, information on some microbiological indicators (faecal streptococci and Clostridium perfringens) was also provided, and their discussion was supplemented in the discussion. Following changes were made in the field of text editing and corrections from the English language, indicated by the reviewers. After the corrections, the text of the manuscript was also checked grammatically by a native speaker. The manuscript co-authors hope that the text will be accepted for publication. We believe that our results certainly contribute to the interest in researching the quality of rainwater in storage tanks. We believe that this is the most desirable subject in times of problems with the management of rainwater and in times of crisis with the availability of water that is safe to use.